# Prognostic Impact and Predictors of New-Onset Atrial Fibrillation in Heart Failure

**DOI:** 10.3390/life12040579

**Published:** 2022-04-13

**Authors:** Hyo-In Choi, Sang Eun Lee, Min-Seok Kim, Hae-Young Lee, Hyun-Jai Cho, Jin Oh Choi, Eun-Seok Jeon, Kyung-Kuk Hwang, Shung Chull Chae, Sang Hong Baek, Seok-Min Kang, Dong-Ju Choi, Byung-Su Yoo, Kye Hun Kim, Myeong-Chan Cho, Byung-Hee Oh, Jae-Joong Kim

**Affiliations:** 1Division of Cardiology, Department of Internal Medicine, Kangbuk Samsung Hospital, Sungkyunkwan University School of Medicine, Seoul 03181, Korea; drhyoin.choi@samsung.com; 2Division of Cardiology, Department of Internal Medicine, Asan Medical Center, University of Ulsan College of Medicine, Seoul 05505, Korea; guess124@gmail.com (M.-S.K.); jjkim@amc.seoul.kr (J.-J.K.); 3Division of Cardiology, Department of Internal Medicine, Seoul National University Hospital, Seoul 03080, Korea; hylee612@snu.ac.kr (H.-Y.L.); hyunjaicho@snu.ac.kr (H.-J.C.); ohbhmed@snu.ac.kr (B.-H.O.); 4Division of Cardiology, Department of Internal Medicine, Sungkyunkwan University College of Medicine, Seoul 06351, Korea; jinoh.choi@samsung.com (J.O.C.); esjeon@skku.edu (E.-S.J.); 5Division of Cardiology, Department of Internal Medicine, Chungbuk National University College of Medicine, Cheongju 28644, Korea; kyungkukhwang@gmail.com (K.-K.H.); mccho@chungbuk.ac.kr (M.-C.C.); 6Division of Cardiology, Department of Internal Medicine, Kyungpook National University College of Medicine, Daegu 41944, Korea; scchae@knu.ac.kr; 7Division of Cardiology, Department of Internal Medicine, The Catholic University of Republic of Korea, Seoul 03083, Korea; whitesh@catholic.ac.kr; 8Division of Cardiology, Department of Internal Medicine, Yonsei University College of Medicine, Seoul 03722, Korea; smkang@yumc.yonsei.ac.kr; 9Division of Cardiology, Cardiovascular Center, Seoul National University Bundang Hospital, Seongnam 13620, Korea; djchoi@snu.ac.kr; 10Division of Cardiology, Department of Internal Medicine, Yonsei University Wonju College of Medicine, Wonju 26426, Korea; yubs@yonsei.ac.kr; 11Heart Research Center, Chonnam National University, Gwangju 61469, Korea; christiankyehun@hanmail.net

**Keywords:** atrial fibrillation, heart failure, mortality, risk factors, registries

## Abstract

Background: The prognostic impact and predictors of NOAF in HF patients are not fully elucidated. This study aims to determine whether new-onset atrial fibrillation (NOAF) affects patient outcome and investigate predictors of atrial fibrillation (AF) in acute heart failure (HF) patients using real-world data. Methods: The factors associated with NOAF in 2894 patients with sinus rhythm (SR) enrolled in the Korean Acute Heart Failure (KorAHF) registry were investigated. Survival was analyzed using AF as a time-dependent covariate. Relevant predictors of NOAF were analyzed using multivariate proportional hazards models. Results: Over 27.4 months, 187 patients developed AF. The median overall survival time was over 48 and 9.9 months for the SR and NOAF groups, respectively. Cox regression analysis with NOAF as a time-dependent covariate showed a higher risk of death among patients with NOAF. Multivariate Cox modeling showed that age, worsening HF, valvular heart disease (VHD), loop diuretics, lower heart rate, larger left atrium (LA) diameter, and elevated creatinine levels were independently associated with NOAF. Risk score indicated the number of independent predictors. The incidence of NOAF was 2.9%, 9.4%, and 21.8% in the low-risk, moderate-risk, and high-risk groups, respectively (*p* < 0.001). Conditional inference tree analysis identified worsening HF, heart rate, age, LA diameter, and VHD as discriminators. Conclusions: NOAF was associated with decreased survival in acute HF patients with SR. Age, worsening HF, VHD, loop diuretics, lower heart rate, larger LA diameter, and elevated creatinine could independently predict NOAF. This may be useful to risk-stratify HF patients at risk for AF.

## 1. Introduction

Atrial fibrillation (AF) is commonly observed in patients with heart failure (HF). In general, the occurrence of AF in patients with HF is known to be associated with a poor prognosis [1,2]. Since AF exerts negative hemodynamic effects by decreasing the cardiac output [3], AF may be associated with worse prognosis in HF patients. In contrast, AF is the most common cause of tachycardia-induced cardiomyopathy, which is generally reversible and shows benign prognosis [4]. The inconsistent prognostic significance of AF may be attributed to the heterogeneous relationship between AF and HF. As active rhythm control for AF is possible [5], it is important to know the effect of AF on the prognosis of patients with HF. In this aspect, we hypothesized that new-onset AF (NOAF) in patients with prior HF might has more clear prognostic effect on HF patients. This has been evaluated in a few studies, and most of them found that NOAF suggests a grave prognosis for HF patients [6].

The early recognition and holistic management of risk factors associated with AF development may help to prevent new AF incidences in patients with HF. The risk stratification for HF patients has become more important since some medications including diuretics, nitrates, ivabradine, and high-dose omega-3 have been associated with an increased incidence of AF, whereas others could help to reduce AF risk [7]. However, studies identifying patients at the highest risk of AF have been inconclusive. Although some scoring methods have been developed to predict the risk of NOAF, none of them are currently recommended or adequately validated as a screening tool, especially for HF patients [8,9]. Previous studies had certain limitations, such as the inclusion of patients with mild symptoms or only patients with reduced ejection fractions, and intrinsic problems such as a retrospective nature or the sub-analysis of randomized trials.

Therefore, we investigated (1) the prognostic impact of NOAF and (2) predictors of NOAF in patients hospitalized with acute HF using a real-world database, the Korean Acute Heart Failure (KorAHF) registry.

## 2. Materials and Methods

We used data from the KorAHF registry, a prospective multicenter cohort study designed to characterize hospitalized patients for acute HF in Korea and their clinical outcomes [10]. Between March 2011 and February 2014, the registry prospectively enrolled 5625 patients who were admitted for acute HF from 10 tertiary university hospitals. Patients with signs or symptoms of HF who met at least one of the following criteria were included: (1) lung congestion; (2) objective findings of left ventricular systolic dysfunction or structural heart disease. Worsening HF was defined as previously diagnosed HF requiring new or augmented pharmacological treatment. Patients with a history of at least moderate valvular disease or valve surgery were classified as having valvular heart disease (VHD). Secondary functional mitral regurgitation and tricuspid regurgitation were excluded from VHD history. Data on follow-up clinical manifestations, biochemical assay results, and medications after discharge were documented at the first follow-up visit at 30 days, again at 3 months, 6 months, and 1 year, and then annually up to 5 years. The attending physician completed a web-based case report form in the Clinical Data Management System (iCReaT) from the Korea National Institute of Health. Mortality data for patients lost to follow-up were determined by the National Insurance Data or National Death Records. The study protocol was approved by the ethics committee at each participating center and was conducted according to the principles of the Declaration of Helsinki. The requirement for written informed consent was waived by the ethics committee at each participating centers.

We enrolled 2894 patients discharged without AF diagnosis or any previous AF history (either paroxysmal and/or persistent). NOAF was defined as the first episode of paroxysmal or persistent AF in patients without AF history at discharge, as documented by resting electrocardiogram (ECG) or 24-h Holter monitoring during the follow-up period. The patients were divided according to NOAF status (NOAF group and sinus rhythm (SR) group).

### Statistical Analysis

Baseline characteristics were summarized using standard descriptive statistics. Continuous variables were presented as mean ± standard deviation, and categorical variables were presented as number and percentage. Differences between continuous variables were compared using Student’s *t*-test or Wilcoxon rank-sum test for independent samples, and differences between categorical variables were analyzed using χ^2^ test or Fisher’s exact test. Univariate Cox proportional hazards models were used to assess the independent effects of each variable on NOAF. A multiple Cox regression model was constructed via a backward selection procedure. Clinical characteristics with a univariate *p* value of <0.1 were included for multiple Cox regression analysis. The performance of the final model was assessed with respect to both discrimination and calibration. Discrimination was quantified with Harrell’s C index, and calibration was assessed using calibration curves. Internal validity was estimated as an optimism-corrected C index with 200 bootstrapping. A risk scoring system was constructed based on the final model. The effect of loop diuretics (B = 0.476) was assigned one point. Other variables were then sequentially assigned points by their β coefficients. AF incidence curves according to the risk group were constructed using the Kaplan–Meier method. In conditional inference tree analysis, variables included in the risk scoring system were also used, and each variable was considered as a potential split. An optimal partitioning was ultimately selected based on multiplicity-adjusted *p* values. A method by Simon and Makuch [11] was used to estimate overall survival curves according to the development of NOAF, which was considered as a time-dependent variable. Comparisons for overall survival were performed using a Cox model with a time-dependent covariate. Differences were considered statistically significant when *p* < 0.05 in two-sided tests. All statistical analyses were performed with SAS 9.4 (SAS Institute, Cary, NC, USA) and R 3.4.2 (www.r-project.org, accessed on 21 June 2021) with packages “rms”, “RcmdrPlugin.EZR”, and “partykit”.

## 3. Results

### 3.1. Patient Characteristics

Among the 5356 patients who were discharged alive, 2424 (45.3%) were previously diagnosed with AF at discharge, whereas 2932 (54.7%) were not. Excluded patients who had previous AF history were older and more likely to have worsening HF, hypertension, VHD, chronic obstructive pulmonary disease, and cerebrovascular accident compared with patients with SR at discharge. The number of prescriptions for HF medications (renin-angiotensin system (RAS) blockers, beta-blockers, mineralocorticoid receptor antagonists (MRAs)) was lower for the excluded patients than the included patients (Appendix A).

Among the 2932 patients without previous AF diagnosis, we excluded 38 patients who died within 7 days after discharge. NOAF was diagnosed in 187/2894 (6.5%) of patients who were followed up for a median period of 27.4 months. Table 1 and Table 2 present the clinical characteristics and laboratory tests according to the presence of NOAF. In comparison with patients without NOAF, patients with NOAF were older and had worsening HF and a higher likelihood of previously diagnosed hypertension and VHD. The rate of ischemic cardiomyopathy did not differ between the two groups. Patients who developed NOAF were more frequently treated with loop diuretics at discharge. The use of RAS blockers was similar for both groups. The use of beta-blockers, aldosterone antagonists, and statins was also similar between the groups. On ECG, a lower heart rate (90.9 beats/min vs. 85.9 beats/min) and a longer PR interval (167.29 msec vs. 179.3 msec) were observed in the NOAF group. Among the echocardiographic parameters, only the left atrium (LA) diameter was significantly different; the LA diameter of patients with NOAF was larger than that of patients with SR (44.96 mm vs. 47.70 mm). The left ventricular ejection fraction did not differ significantly. The median B-type natriuretic peptide (BNP) and N-terminal pro-B-type natriuretic peptide (NT-proBNP) levels at discharge did not show significant differences.

### 3.2. Cox Regression Analysis for NOAF and Score Development

The predictive value of each variable was analyzed using Cox proportional hazards models (Table 3). In a multivariate analysis, older age, worsening HF, history of VHD, use of loop diuretics, larger LA diameter, lower heart rate at discharge, and higher creatinine level at discharge were significantly associated with AF development. Model discrimination was modest for the development of AF (C index 0.685, 95% CI 0.641–0.729). Internal validation was performed using 200 bootstrap resamples. The optimism-corrected C index was 0.674 (95% CI 0.630–0.718). The risk score was defined as the number of independent predictors. History of VHD and worsening chronic HF were included as predictors. In addition, the risks of continuous variables were classified according to their mean values (65 years for age, 45 mm for LA diameter, 77 beats/min for heart rate at discharge, and 1.5 mg/dL for creatinine level at discharge). We defined patients with scores of 0–2, 3–4, and over 5 as having a low, intermediate, and high risk of NOAF, respectively. The incidence of NOAF was 2.9% in the low-risk group, 9.4% in the moderate-risk group, and 21.8% in the high-risk group (*p* < 0.001) (Figure 1).

### 3.3. Survival Tree Analysis for NOAF

In conditional inference tree analysis (Figure 2), worsening HF was identified as the best discriminator for future AF development (*p* < 0.001). If patients fell into the “no” category of worsening HF, the heart rate at discharge provided additional diagnostic value. Among patients with worsening HF, age was the next discriminator after the HF category. Among patients >56 years old, LA diameter was a useful discriminator. Among patients whose LA diameter was <56 mm, previous VHD history was a useful discriminator. The accuracy of the decision tree was 67.8% (63.6–72.1). AF incidence was significantly higher in those who had multiple predictors than in those who had few (Figure 3).

### 3.4. Prognostic Impact of NOAF on Patient Survival

Overall, 942 (32.6%) patients died during a median follow-up period of 27.4 months after hospital discharge. Figure 4 shows the survival curves with respect to AF status as a time-dependent covariate. Patients with NOAF had the worst prognosis (red line), and patients without AF had the best prognosis (black line). The gray line indicates the survival curve for patients with AF history (excluded patients). The median overall survival time was over 48 months for the SR group and 9.9 months for the NOAF group. In the SR group, the median survival was not reached during the follow-up period. Cox regression analysis with NOAF as a time-dependent covariate revealed a higher risk of death (5.5 times) for patients who developed NOAF (HR 5.54, 95% CI 4.28–7.16, *p* < 0.001). In comparison with patients with previous AF (excluded patients), patients with SR had slightly better survival (HR 1.2, 95% CI 1.1–1.3, *p* < 0.001).

## 4. Discussion

In the present study, (1) 6.5% of HF patients developed AF during a follow-up period of 27.4 months; (2) patients who developed AF had worse prognosis than patients with a history of AF or SR; (3) older age, VHD, loop diuretics, larger LA diameter, low heart rate, and increased creatinine levels were independently associated with NOAF development in multivariate analysis; and (4) patients with multiple predictors of AF had a 7.5-fold higher risk of developing AF compared with patients without any of these predictors. These findings are clinically important because an understanding of the effect and predictors of NOAF is crucial for the better management of HF patients.

In this study, 187 of the 2894 patients without AF history (6.5%) developed AF during a median follow-up period of 27.4 months with an annual incidence of approximately 30.5 per 1000 person/year. The incidence rate of NOAF was consistent with the findings of previous studies, including the prospective comparison of ARNI with ACEI to Determine Impact on Global Mortality and morbidity in Heart Failure (PARADIGM-HF) and ATMOSPHERE (Aliskiren Trial to Minimize Outcomes in Patients with Heart Failure) trials [6], which showed a 3–5% annual incidence of AF in HF patients.

An interesting finding of our study is the association between future AF and low heart rate at discharge. An elevated resting heart rate has been established as an independent risk factor for cardiovascular mortality and morbidity [12]. However, the association between heart rate and incident AF is less clear. Recent studies have shown that a low resting heart rate may be independently associated with increased AF risk [13,14]. These findings suggest that underlying alterations in autonomic tone and subclinical sinus node dysfunction exist in individuals with a low heart rate, which could predispose them to AF. Patients with a low resting heart rate may have subclinical sinus node dysfunction [15], possibly indicating an underlying atrial disease that may contribute to AF predisposition. Additionally, a low resting heart rate may be associated with increased vagal tone, which predisposes a patient to AF [16]. Additional research is required to determine the optimal heart rate threshold for patients with AF and HF.

We did not observe a significant reduction in NOAF in patients who were treated with angiotensin-converting enzyme inhibitors (ACEIs) or angiotensin II receptor blockers (ARBs). Previous studies have reached divergent conclusions about the efficacy of RAS inhibition for the primary prevention of AF. Although some studies have shown a reduction in new-onset events with either ACEI or ARB therapy [17], others have not demonstrated a clear benefit [18]. Furthermore, previous studies have shown that MRAs [19,20], beta-blockers [21], and statins [22] could significantly reduce the risk of NOAF in patients with HF history. In contemporary clinical practice, most HF patients are likely to have already received HF medications including RAS blockers, MRAs, beta-blockers, and statins. In our study, the majority (72%) of patients not treated with RAS blockers used one of the other drug groups—MRAs, beta-blockers, and statins; this may have affected our results. Insufficiency of optimal HF medication is a concern that must be addressed in the future.

The present study demonstrated that diuretics were independently associated with increased NOAF risk in patients with HF. Other studies have demonstrated an association between diuretic use and NOAF [23]. The occurrence of hypokalemia may explain the association between diuretic therapy and NOAF; diuretic therapy may induce hypokalemia and facilitate atrial beats by re-entry or automatic mechanisms [24]. A possible mechanism leading to this proarrhythmic effect includes the alteration of ionic channel activity in atrial myocytes by diuretics. Although this alteration cannot be detected by measuring serum potassium levels, it can exert a proarrhythmic effect on an abnormal atrial substrate. Alternatively, loop diuretics, by stimulating the renin-angiotensin-aldosterone system, may favor AF recurrence.

In the present study, LA enlargement was found to be the only echocardiographic predictor for NOAF in HF patients. The association of LA enlargement with AF development has been reported in previous studies [8,25]. LA dilation itself can be a trigger of AF; however, it can also be the result of long-term left ventricular compliance or relaxation abnormality. Interestingly, the present study showed that a relatively mild enlargement of the LA (≥45 mm) was associated with increased AF risk. However, the Warfarin versus Aspirin in Reduced Cardiac Ejection Fraction (WARCEF) trial found that a mild enlargement of LA diameter (>45 mm) was a significant predictor for AF recurrence in a multivariate model of patients with HF [8]. LA diameter provides useful information to help clinicians identify patients with a high risk of NOAF, allowing closer monitoring or earlier intervention for these patients.

We provide a scoring system based on clinical and laboratory variables that may easily be used in primary care to predict an individual’s risk of developing AF. For the classification of individuals into risk groups, the risk score yielded reasonable accuracy. This multivariate risk scoring approach may identify persons with HF who have a high risk of AF and could benefit from ECG monitoring and aggressive control of correctable predisposing variables.

In our study, patients with NOAF had a significantly decreased survival compared to patients with SR or previous AF. The median survival time after NOAF development was only 9.9 months, and there was a higher risk of all-cause death (5.5 times). Based on the Framingham Heart Study cohort, Wang et al., reported that NOAF was associated with a significantly increased risk of all-cause death; however, previous AF was not associated with an increased risk of mortality in aging patients with severely decompensated acute heart failure [26]. In comparison with NOAF, previous AF had a minimal effect on all-cause death in the present study. As tachycardia-induced cardiomyopathy is a common cause of the HF admission of acute HF patients with previous AF, this may be associated with a better prognosis.

This study had certain limitations. Cases of AF were identified predominately using ECG data and hospital discharge records. Potentially, non-permanent AF cases were missed because of the intermittent nature of AF, representing one of the major shortcomings of our study. In addition, as the KorAHF registry reflects real-world management, concomitant treatment was left to physician’s discretion; as a consequence, any apparent effects due to other drugs, such as diuretics, antiarrhythmics, beta-blockers, ACEIs, and statins, during AF occurrences could have caused bias. Finally, we developed and validated a risk scoring system with an Asian cohort of acute HF with heterogeneous disease factors; thus, our scoring system may not be extrapolated to other ethnic groups or a single disease entity.

## 5. Conclusions

In conclusion, NOAF may be associated with decreased survival in HF patients with SR. Age, VHD, loop diuretics, lower heart rate, and elevated creatinine levels could independently predict NOAF. This information may be useful to risk-stratify HF patients for NOAF development, allowing close monitoring and possibly early detection.

## Figures and Tables

**Figure 1 life-12-00579-f001:**
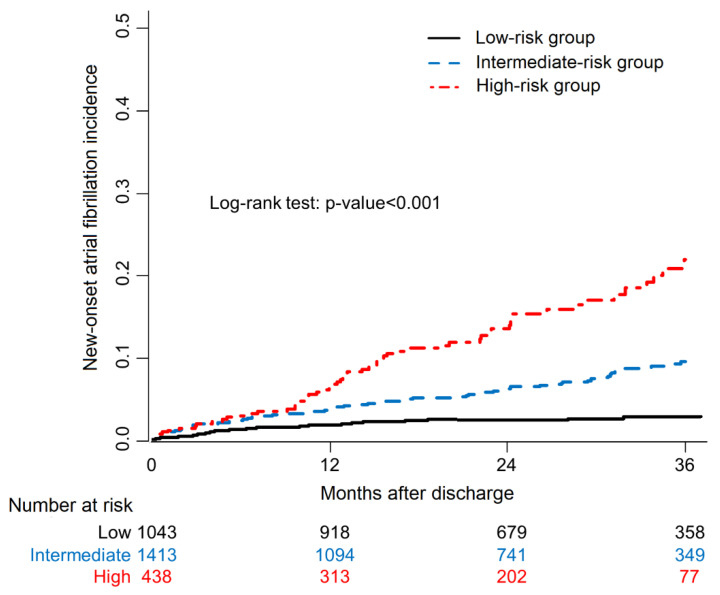
Product-limit survival estimate curves for new-onset atrial fibrillation incidence. Patients with multiple risk factors had significantly higher incidences of new-onset atrial fibrillation.

**Figure 2 life-12-00579-f002:**
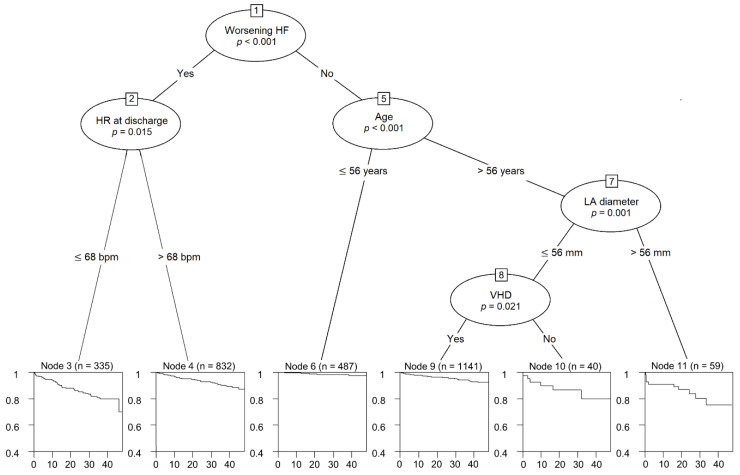
Conditional inference trees for worsening heart failure, heart rate at discharge (beat/min), age (years), left atrial diameter (mm), and valvular heart disease as predictors of new-onset atrial fibrillation. HF, heart failure; HR, heart rate; LA, left atrial; VHD, valvular heart disease.

**Figure 3 life-12-00579-f003:**
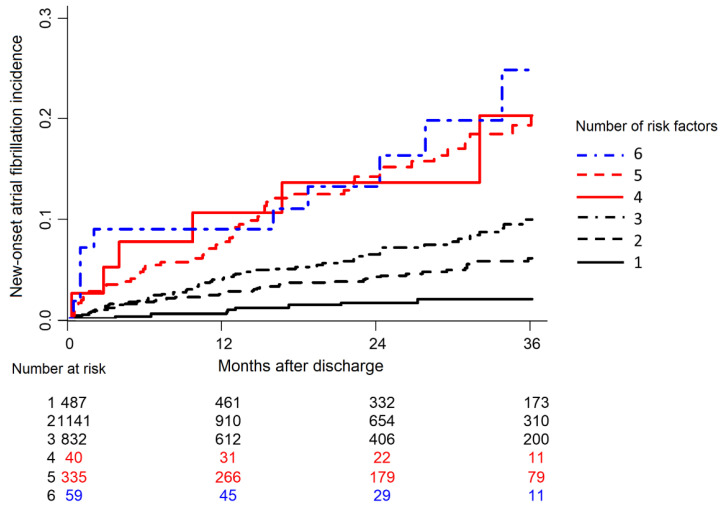
New-onset atrial fibrillation incidence curves according to the presence of predictors. Patients with multiple risk factors had the highest incidence rate of new-onset atrial fibrillation.

**Figure 4 life-12-00579-f004:**
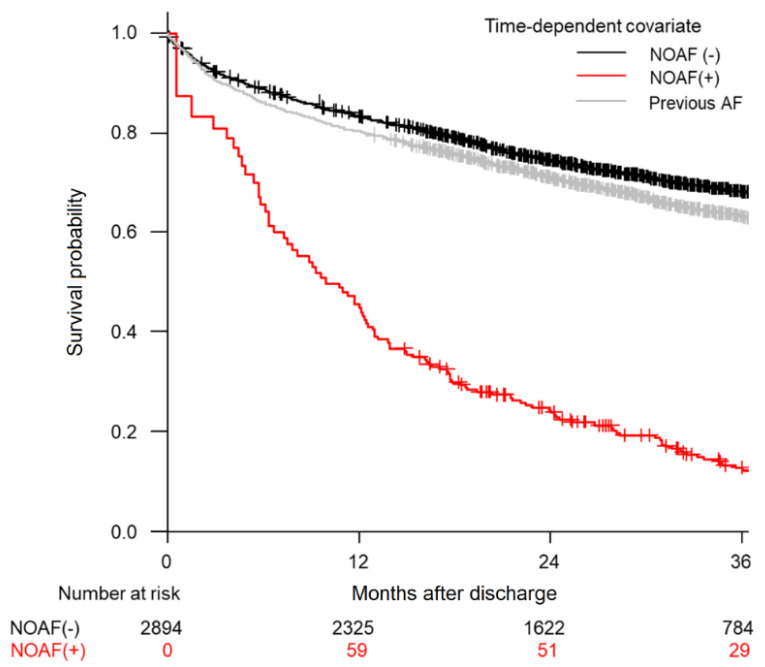
Survival curves with respect to atrial fibrillation status as a time-dependent covariate. Patients with new-onset atrial fibrillation had the worst mortality compared to previous atrial fibrillation or sinus rhythm groups.

**Table 1 life-12-00579-t001:** Baseline and discharge characteristics according to NOAF status.

	SR Group (N = 2707)	NOAF Group (N = 187)	*p* Value
Age, year	66.2 ± 15.5	70.8 ± 13.2	<0.001
Male (%).	1475 (54.5)	92 (49.2)	0.18
BMI, kg/m^2^	23.4 ± 3.93	23.8 ± 3.65	0.19
Current smoking (%)	588 (21.7)	27 (14.4)	0.01
Hypertension (%)	1655 (61.1)	133 (71.1)	0.008
Diabetes mellitus (%)	1159 (42.8)	89 (47.6)	0.23
Chronic kidney disease (%)	397 (14.7)	36 (19.3)	0.11
Chronic obstructive lung disease (%)	266 (9.8)	20 (10.7)	0.80
Cerebrovascular accident (%)	353 (13.0)	32 (17.1)	0.14
Implantable cardioverter defibrillator (%)	73 (2.7)	7 (3.7)	0.54
Cardiac resynchronization therapy (%)	46 (1.7)	4 (2.1)	0.88
Percutaneous coronary intervention (%)	527 (19.5)	49 (26.2)	0.03
Coronary artery bypass surgery (%)	145 (5.4)	17 (9.1)	0.04
Ischemic heart disease (%)	1387 (51.2)	98 (52.4)	0.82
Dilated cardiomyopathy (%)	214 (7.9)	23 (12.3)	0.05
Valvular heart disease (%)	195 (7.2)	29 (15.5)	<0.001
Heart failure category			<0.001
De novo heart failure	1648 (60.9)	79 (42.2)	
Acute decompensated heart failure	1059 (39.1)	108 (57.8)	
Systolic BP, mmHg	134.0 ± 31.3	133.2 ± 31.11	0.76
Diastolic BP, mmHg	79.4 ± 18.8	78.6 ± 17.1	0.56
Heart rate, beats/min	91.0 ± 22.2	85.6 ± 22.8	0.001
NYHA class (%)			0.28
II	459 (17.0)	24 (12.8)	
III	952 (35.2)	73 (39.0)	
IV	1296 (47.9)	90 (48.1)	
Discharge medication			
RAS blocker (%)	1971 (72.8)	129 (69.0)	0.29
Beta-blocker (%)	1490 (55.0)	100 (53.5)	0.73
Aldosterone antagonist (%)	1215 (44.9)	79 (42.2)	0.53
Nitrates (%)	695 (25.7)	54 (28.9)	0.38
Loop diuretics (%)	1887 (69.7)	148 (79.1)	0.008
Thiazide (%)	206 (7.6)	12 (6.4)	0.65
Statin (%)	1342 (49.6)	92 (49.2)	0.98
Clinical status on discharge			
BMI, kg/m^2^	22.5 ± 3.8	22.8 ± 3.7	0.39
Systolic BP, mmHg	115.2 ± 18.4	115.6 ± 19.6	0.78
Diastolic BP, mmHg	67.3 ± 11.6	65.3 ± 10.3	0.02
Heart rate, beats/min	77.6 ± 13.4	73.5 ± 13.2	<0.001
NYHA class (%)			0.26
I	522 (20.0)	29 (16.1)	
II	1786 (68.5)	136 (75.6)	
III	178 (6.8)	9 (5.0)	
IV	121 (4.6)	6 (3.3)	

NOAF, new-onset atrial fibrillation; BMI, body mass index; BP, blood pressure; NYHA, New York Heart Association; BUN, blood urea nitrogen; BNP, blood natriuretic peptide; NT-proBNP, N-terminal proBNP; RBBB, right bundle branch block; LBBB, left bundle branch block; LV, left ventricular; EF, ejection fraction; LA, left atrium; TR, tricuspid regurgitation; RV, right ventricular; RAS, renin-angiotensin system; SR, sinus rhythm.

**Table 2 life-12-00579-t002:** Laboratory, electrocardiographic and echocardiographic results according to NOAF status.

	SR Group (N = 2707)	NOAF Group (N = 187)	*p* Value
Laboratory tests at admission			
Sodium, mmol/L	137.7 ± 4.5	137.3 ± 4.3	0.17
Creatinine, mg/dL	1.5 ± 1.6	1.7 ± 1.7	0.12
BUN, mg/dL	25.3 ± 16.0	28.6 ± 17.3	0.006
BNP, pg/mL	1445 ± 1335	1844 ± 1624	0.02
NT-proBNP, pg/mL	9208 ± 10,875	11,474 ± 16,883	0.05
Electrocardiographic parameters			
RBBB (%)	171 (6.3)	8 (4.3)	0.34
LBBB (%)	186 (6.9)	13 (7.0)	1.00
Interventricular conduction disturbance (%)	129 (4.8)	10 (5.4)	0.84
Q wave (%)	392 (14.5)	30 (16.1)	0.61
PR interval, msec	167.3 ± 32.9	179.3 ± 49.5	<0.001
QRS width, msec	107.1 ± 28.6	110.0 ± 31.1	0.18
QT interval, msec	396.2 ± 58.9	408.7 ± 68.3	0.006
Corrected QT interval, msec	476.2 ± 43.7	475.8 ± 49.2	0.90
Heart rate, beats/min	90.9 ± 22.4	85.9 ± 23.1	0.003
Echocardiographic parameters			
LV end-diastolic dimension, mm	58.3 ± 10.2	59.0 ± 10.2	0.37
LV end-systolic dimension, mm	46.5 ± 12.5	46.5 ± 12.4	0.999
LV EF, %	36.3 ± 15.2	36.8 ± 16.0	0.63
LA diameter, mm	44.9 ± 8.2	47.7 ± 8.4	<0.001
Peak TR velocity, m/sec	2.9 ± 0.6	2.9 ± 0.6	0.94
Estimated RV systolic pressure, mmHg	43.3 ± 15.6	45.9 ± 17.1	0.07
Laboratory tests at discharge			
Sodium, mmol/L	138.0 ± 3.8	137.8 ± 3.7	0.49
Creatinine, mg/dL	1.4 ± 1.5	1.7 ± 1.8	0.03
BUN, mg/dL	23.8 ± 14.9	27.3 ± 18.3	0.002
BNP, pg/mL	1261 ± 1277	1605 ± 1440	0.03
NT-proBNP, pg/mL	8555 ± 10,668	10,426 ± 16,667	0.10

NOAF, new-onset atrial fibrillation; BMI, body mass index; BP, blood pressure; NYHA, New York Heart Association; BUN, blood urea nitrogen; BNP, blood natriuretic peptide; NT-proBNP, N-terminal proBNP; RBBB, right bundle branch block; LBBB, left bundle branch block; LV, left ventricular; EF, ejection fraction; LA, left atrium; TR, tricuspid regurgitation; RV, right ventricular; RAS, renin-angiotensin system; SR, sinus rhythm.

**Table 3 life-12-00579-t003:** Multivariable Cox proportional hazards model (N = 2894, NOAF = 187) for NOAF.

	Multivariable			
Variable	HR	95% CI		*p* Value
Age	1.03	1.01	1.04	<0.001
Worsening heart failure	1.66	1.22	2.24	0.001
Valvular heart disease	1.67	1.11	2.51	0.01
LA diameter	1.03	1.01	1.05	<0.001
Use of loop diuretics at discharge	1.61	1.12	2.32	0.01
Heart rate at discharge	0.98	0.97	1.00	0.01
Creatinine	1.11	1.03	1.20	0.01

NOAF, new-onset atrial fibrillation; HR, hazard ratio; CI, confidence interval; LA, left atrium.

## Data Availability

The data presented in this study are available on request from the corresponding author.

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
