# Peer review of "Prognostic Impact and Predictors of New-Onset Atrial Fibrillation in Heart Failure"

_life, 2022, doi:10.3390/life12040579_

Round 1

Reviewer 1 Report

I have read with great interest the manuscript entitled “Prognostic Impact and Predictors of New-Onset Atrial 2 Fibrillation in Heart Failure.”

I would like to congratulate the authors for their project. Briefly, this study aimed to investigate 1) the prognostic impact of NOAF and 2) predictors of NOAF in patients hospitalized with acute HF using a real-world database, the Korean Acute Heart Failure (KorAHF) registry.

Comments

  1. Line 54: AF has been strongly correlated with a worse prognosis for HF patients (ie 10.1007/s10741-021-10133-6). Authors should add relevant literature and correct it.
  2. Figure 2,3 and 4 should be significantly improved.
  3. It would be interesting to add information about patients having undergone PCI or CABG.

Reviewer 2 Report

I read the submitted manuscript with great interest. The authors raised a fundamental issue that may have a tangible impact on managing patients with heart failure. The great advantage of the study is the relatively large study population. The authors pay attention to the limitations of the work. Nevertheless, the presented study provides valuable insight into the nature of atrial fibrillation associated with heart failure. Therefore, I have no major reservations about accepting the article for publication, apart from minor comments and doubts.
Patients who developed atrial fibrillation (6.5%) were older and at risk of a more severe clinical course due to multiple morbidities. In addition, lower heart rate and conduction disturbances in the form of a prolonged PQ were factors that promoted the occurrence of atrial fibrillation. On the other hand, pharmacological slowing of the heart rate is one of the treatment strategies for patients with heart failure. Therefore, can it be assumed that there is a heart rate threshold below which it should not be lowered? 
I was surprised to find that patients who did not develop AF were more likely to be addicted to tobacco.
Only a borderline difference in the concentration of NTproBNP between both groups was present, without statistical significance.
Why did the authors use the mean value of the continuous variables to construct the risk scale? What made this choice? It would seem that the median or value identified in the ROC curve analysis could be a better choice.
Are the data in Table 3 derived from a single multivariate regression model or multiple models?
The great value of the work is the inclusion of conditional inference trees for worsening HF in it, which organizes the results described and constitutes a specific algorithm that is easy for practical use.
Interestingly, patients with a history of prior AF had a worse prognosis than those with a recent diagnosis. It seems unlikely that all patients with previous AF will have to recover from tachyarrhythmic cardiomyopathy.
Surprisingly, only 70% of patients received drugs that block the renin-angiotensin system.
